# The Microflora of Maize Grains as a Biological Barrier against the Late Wilt Causal Agent, *Magnaporthiopsis maydis*

**Ofir Degani** [1,2,*] **, Danielle Regev** [1,2] **and Shlomit Dor** [1,2]

1   Plant Sciences Department, MIGAL–Galilee Research Institute, 2 Tarshish St., Kiryat Shmona 11016, Israel; linkar45@gmail.com (D.R.); dorshlomit@gmail.com (S.D.)
2   Faculty of Sciences, Tel-Hai College, Upper Galilee, Tel-Hai 12210, Israel
*   Correspondence: d-ofir@bezeqint.net or ofird@telhai.ac.il; Tel.: +972-54-6780114

**Abstract:** The maize pathogen *Magnaporthiopsis maydis* causes severe damage to commercial fields in the late growth stages. This late wilt disease has spread since its discovery (the 1980s) and is now common in most corn-growing areas in Israel. In some fields and sensitive plant species, the disease can affect 100% of the plants. The *M. maydis* pathogen has a hidden endophytic lifecycle (developed inside the plants with no visible symptoms) in resistant corn plants and secondary hosts, such as green foxtail and cotton. As such, it may also be opportunist and attack the host in exceptional cases when conditions encourage it. This work aims to study the pathogen's interactions with maize endophytes (which may play a part in the plant's resistance factors). For this purpose, 11 fungal and bacterial endophytes were isolated from six sweet and fodder corn cultivars with varying susceptibility to late wilt disease. Of these, five endophytes (four species of fungi and one species of bacteria) were selected based on their ability to repress the pathogen in a plate confrontation test. The selected isolates were applied in seed inoculation and tested in pots in a growth room with the Prelude maize cultivar (a late wilt-sensitive maize hybrid) infected with the *M. maydis* pathogen. This assay was accompanied by real-time qPCR that enables tracking the pathogen DNA inside the host roots. After 42 days, two of the endophytes, the *Trichoderma asperellum*, and *Chaetomium subaffine* fungi, significantly ($p < 0.05$) improved the infected plants' growth indices. The fungal species *T. asperellum*, *Chaetomium cochliodes*, *Penicillium citrinum*, and the bacteria *Bacillus subtilis* treatments were able to reduce the *M. maydis* DNA in the host plant's roots. Studying the maize endophytes' role in restricting the invasion and devastating impact of *M. maydis* is an essential initial step towards developing new measures to control the disease. Such an environmentally friendly control interface will be based on strengthening the plants' microbiome.

**Keywords:** *Cephalosporium maydis*; crop protection; endophytes; fungus; *Harpophora maydis*; microbiome; pots assay; real-time PCR; *Trichoderma asperellum*

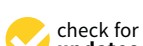



## 1. Introduction

The pathogen *Magnaporthiopsis maydis* (former names *Cephalosporium maydis* and *Harpophora maydis* [1,2]) causes severe damage to cornfields in the late growth stages (near the time of harvest). This late wilt disease (LWD) has spread since its discovery in Egypt in the 1960s. It is now common in at least eight countries, including India [3], Hungary [4], Romania [5], Spain and Portugal [6], Israel [7], and Nepal [8]. In Egypt [9] and Israel [10], the disease is now reported in most maize-growing areas and is considered one of the most devastating threats in this crop. In some fields and sensitive plant species, the disease can affect 100% of the plants [9,11]. LWD is characterized by a relatively fast wilting of the corn that usually occurs at the age of 60–80 days, from before the flowering stage (tasseling) to physiological maturation. The first signs of dehydration, which may appear 50 days after sowing (DAS), progress from the lower part of the plant upwards and eventually cause dehydration and yield loss [7,12]. The pathogen can survive and spread through infected

soils [13], crop residues [14], infected seeds [15], and alternative hosts such as *Lupinus termis* L. (lupine) [16] and watermelon [17]. In LWD-resistant corn plants and secondary hosts, such as *Setaria viridis* (common names green foxtail, green bristlegrass, and wild foxtail millet) and cotton [17,18], *M. maydis* can live inside the plant with no visible symptoms. As such, it may become pathogenic and cause disease in the right conditions. A recent focused research effort has yielded success, and we have an effective and economical chemical pesticide method for dealing with the disease [10,19,20]. Still, the technique's application requires changes in the growing practice and dripline irrigation, and there is a constant threat of resistance development against the preparation. Moreover, chemical approaches have unwanted widespread environmental effects and hazard risks, and they can influence beneficial microorganisms in the soil [21].

Today, the primary measure of dealing with LWD is the cultivation of disease-resistant corn genotypes. This strategy is environmentally friendly and cost-effective to implement but requires constant effort to scan and identify new varieties. That is because prolonged growth of a resistant maize strain (for several years) may eventually result in the selection and outbreak of the pathogen's virulent lines capable of causing disease of this strain, as reported in Egypt and Spain [22–24]. This scenario occurred in Israel to the LWD-resistant corn cultivar, Royalty, which became the leading corn genotype during the late wilt disease outbreak in the 1990s [12,20]. Moreover, most resistant hybrids to LWD are low-yielding or have other undesirable agronomic characteristics [25]. A program to develop new hybrid strains, resistant to LWD, has been operating in Egypt since the 1980s [26], in Israel for more than a decade (R&D North, Migal–Galilee Research Institute, Kiryat Shmona, Israel) [10] and was also reported in India [27].

One of the research directions designed to address these challenges is biological control [28–32]. These methods include manipulating and strengthening beneficial microorganisms communities in the soil (for example, adding compost [29]) or direct applications of antagonistic bacteria and fungi or their secreted product. Such bacteria are *Bacillus subtilis* MF497446 and *Pseudomonas koreensis* (plant growth-promoting rhizobacteria [32]), and mixed strains of cyanobacteria known as *Anabaena oryzae*, *Nostocmuscorum*, and *N. calcicolawere* [29]. Instead, another approach was to use marine algae, and the cyanobacteria *Anabaena oryzae* extracts antifungal activates to targeting the LWD pathogen. These algae include *Jania rubens*, *Corallina elongata*, *Laurencia obtusa*, *Gelidium crinale*, *Enteromorpha compressa*, and *Ulva fasciata*.

LWD biological control using *Trichoderma* spp. has also been demonstrated. The species in this genus can form mutualistic endophytic relationships with several plant species [33], while other species have been developed as biocontrol agents against fungal phytopathogens [34]. Previously, *T. cutaneum* reduced the incidence of *M. maydis* LWD of maize under greenhouse conditions by 89% compared to the control (from 94% to 11%) [31]. Similarly, *T. harzianum* applied in the field reduced the pre-emergence damping-off from 47% to 32% and increased the survival plants by 59% [35]. The use of *T. viride* alone, or even better with chitosan NPs, resulted in controlling late wilt in the greenhouse and field trials and improving the plants' growth parameters [36]. Combining *T. viride* with the mycorrhizae led to a law effect on disease control.

It was also shown that microalgae, *Chlorella vulgaris* extracts, with each of the *Trichoderma* species, *T. virens*, and *T. koningii*, were effective treatments against LWD under greenhouse and field conditions [28]. The potential for applying *Trichoderma*-based methods against *M. maydis* in Israel has only lately been tested against the Israeli pathogen strains [37]. Examining eight marine and soil isolates of *Trichoderma* spp., known for their high mycoparasitic potential, revealed that *T. longibrachiatum* isolates and *T. asperelloides* has strong antagonistic activity against the Israeli pathogen isolate. These bioprotective agents were tested in a series of experiments in the laboratory and a growth room under controlled conditions until their final examination in pots under field conditions throughout a full growing season [37]. This green treatment has significantly improved the growth and yield indices to healthy plant levels and reduced pathogen DNA in the plant tissues by 98%.

Despite this continuous effort, little or no information exists in the literature on the role of the plant's microbiome regarding its immunity to LWD. Endophytes are mutualistic symbionts within healthy plant tissues. Maize endophytes can be either acquired from the environment (horizontal transmission) or inherited via seed (vertical transmission) [38]. Of the ~300 maize-associated endophytes, over 90% belong to bacteria, either *Firmicutes* or *Proteobacteria*, with the genus *Bacillus* by far the most common, followed by *Burkholderia*, *Enterobacteria*, and *Paenibacillus* [38]. Among the fungal endophytes, the most common genera, include known plant pathogens, such as *Alternaria*, *Fusarium*, and Acremonium, and known beneficial organisms such as *Trichoderma* [38]. Much research has focused on identifying endophytes that antagonize pathogens for their potential use in biocontrol [39]. Several species have broad antipathogen properties so that a single endophyte can protect against many diseases [38]. Other protective effects are indirect, for instance, the induction of systemic plant resistance by *Trichoderma*.

Indeed, endophytic fungi were previously studied for their protective effect against *Fusarium verticillioides* and *Cladosporium herbarum*, the causal agents of maize seedling blight, and stalk and root rot. The endophytes *Trichoderma koningii* and *Alternaria alternata* showed promising biopotential ability in confronting (antagonism) assays [40]. Bacterial endophytes, such as *Bacillus amyloliquifaciens* or *Bacillus subtilis* naturally occur in many maize varieties. Their study suggests that they may function to protect hosts by secreting antifungal lipopeptides. This secreted compound inhibits pathogens and induces host plant pathogenesis-related genes' upregulation (systemic acquired resistance) [41].

Despite considerable research efforts in recent decades, achieving new and effective environmental-friendly ways to control LWD is a continuous goal. Specifically, as far as we know, the role of endophytes in maize plants' ability to resist LWD is yet to be discovered. In this study, we hypothesize that endophytic fungi and bacteria communities that are natural inhabitants of different maize varieties grains may impact plant health and could be an essential contribution to sustainable integrated agriculture. To test this hypothesis, we isolated endophytic fungi and bacteria from several varieties of maize and evaluated their capabilities to inhibit *M. maydis* in direct-confront assays on media plates. The successful microorganisms capable of inhibiting pathogenic growth by direct contact or producing secreted antifungal compounds to the medium were selected and tested in seed treatment in sprouts.

## 2. Materials and Methods

### 2.1. Origin and Growth of the Maize Pathogen, M. maydis

One representative *M. maydis* isolate designated Hm-2 (CBS 133165, deposited in the CBS-KNAW Fungal Biodiversity Center, Utrecht, The Netherlands) was selected for this study from our isolates library. This isolate was previously recovered in 2001 from a cornfield on Kibbutz Sde Nehemia in the Hula Valley in Upper Galilee, northern Israel, from Jubilee cv. corn plants showing dehydration symptoms. The Israeli *M. maydis* isolates were examined, characterized, and identified as previously described [7]. The isolate was grown on potato dextrose agar (PDA, Difco Laboratories Detroit, Detroit, MI, USA). Transferring the fungus to a new plate was done by cutting a 6-mm (in diameter) colony agar disk from the margins of *M. maydis* culture and sowing it to a new PDA petri dish. Colonies were grown for 4–6 days in a 28 ± 1 °C incubator in the dark.

### 2.2. Preparation of Maize Grains

Prior to use for the isolation of endophytes or in the pot pathogenicity assay, maize seeds were treated in several preparation steps. The corn seeds were commercial and were provided courtesy of the seed companies listed in Table 1. All seeds were pretreated by the seed companies with Captan (cis-N-trichloromethylthio-4-cyclohexene-l,2-dicarboximide), the most common fungicide for maize seed dressing in Israel. The seeds were washed with tap water and stirred 4–5 times, while changing the water until rinsing the Captan. They were then externally disinfected by soaking in 1% sodium hypochlorite (NaOCl) for

1 min and rinsed twice in sterile double-distilled water (DDW). The seeds were dried in a biological hood on sterile paper.

**Table 1.** Maize cultivars tested for endophytes presence.

| Cultivar | Type | Seed Company | Supply Company | Degree of LWD Sensitivity |
|---|---|---|---|---|
| Simon | Fodder | Semillas Fitó, Barcelona, Spain | Tarsis Inc., Petach Tikva, Israel | Highly resistant |
| Hatai | Fodder | Semillas Fitó, Barcelona, Spain | Tarsis Inc., Petach Tikva, Israel | Highly resistant |
| Megaton | Sweet | Zeraim Gedera-Syngenta, Kibbutz, Revadim, Israel | Hazera Seeds Ltd., Berurim MP Shikmim, Israel | Hypersensitive |
| Prelude | Sweet | SRS Snowy River Seeds, Australia | Green 2000 Ltd., Israel | Sensitive |
| Jubilee | Sweet | Pop Vriend Seeds B.V., Andijk, The Netherlands | Eden Seeds, Reut, Israel | Sensitive |
| Royalty | Sweet | Pop Vriend Seeds B.V., Andijk, The Netherlands | Eden Seeds, Reut, Israel | Resistant |

*2.3. Isolation and Identification of Endophytes from Maize Grains*

In the study, 11 endophytes were isolated from four sweet and two fodder corn varieties with varying susceptibility to LWD (Table 1). Each seed was cut lengthwise using a sterile scalpel and placed on a potato dextrose agar (PDA) substrate with the cutting surface turned downwards (10 half-seeds per 90 mm plate). Petri plates were incubated at $28 \pm 1$ °C in the dark for 2–3 days. At the end of this period, the fungal and bacterial endophytes that developed were isolated into new PDA plates and grown at $28 \pm 1$ °C in the dark for another 5–6 days. After that, they were subjected to molecular identification.

*2.4. Plate Confrontation Assay*

The ability for selected endophytes to restrict *M. maydis* growth by direct hyphae contact, secretion of antifungal compounds, or growth above its colony surface is the first step in revealing their biocontrol potential. To this end, a confrontation test (antagonism or mycoparasitism) was performed, as previously described [18], by placing colony agar disks on a 90-mm-diameter rich medium petri dish. Agar disks (6 mm in diameter) containing endophytes were cut from the culture's margins and placed on a PDA plate in front of a similar disk taken from *M. maydis* culture margins. Dishes were labeled and incubated at $28 \pm 1$ °C in the dark. The interactions between the *M. maydis* pathogen and each of the endophyte isolates were documented and photographed after 6–7 days. Endophytes that managed to restrict the pathogen's growth or grew on the pathogen mycelium were marked as having microparasitic potential. Each endophyte was tested in five independent repeats, and similar results were obtained. One representative plate was selected and is presented in the Results section.

*2.5. Pathogenicity Assay in Sprouts*

The sprouts test examined the outcome of maize seeds' enrichment with selected endophytes, in preventing LWD symptoms at the early growth period. The experiment was performed in a growth room (16 h of light, average temperature 25 °C, about 30% humidity). Pots with a volume of 2 L of heavy local peat soil were prepared. The soil was from the Gadash Amir field (Mehogi-1 plot, coordinates: 33°09′59″ N 35°36′52″ E, Hula Valley in Upper Galilee, northern Israel), having a history of pathogen infestations [10,11]. The soil was mixed with 30% Perlite No. 4 for ground aeration. The inoculation method also consists of sterilized, and *M. maydis* colonized wheat grains that were used here to disperse the pathogen in the soil, as described previously [23]. To this end, wheat seeds were allowed to swell for 4 h in tap water, were autoclave-sterilized, and then incubated at $28 \pm 1$ °C with *M. maydis* for about 10 days by adding five mycelial discs per bottle containing 50 cm$^3$ of sterilized seeds. The mycelial discs were cut from PDA colony margins, grown as described in Section 2.1.

Corn seeds were enriched with five selected endophytes and tested against control plants without enrichment. Prelude cv. (sweet maize from SRS Snowy River Seeds, Australia, supplied by Green 2000 Ltd., Bitan Aharon, Israel), susceptible to late wilt

disease, were incubated with selected endophytes before sowing, as follows. In separate 250 mL Erlenmeyer flasks (three per endophyte), 10 maize seeds and six endophyte colony discs (6 mm in diameter) were incubated for two days in moderate humidity (5 mL DDW in each Erlenmeyer) at $28 \pm 1$ °C in the dark. The control seeds underwent similar incubation without endophytes. In each pot, five seeds were sown to a depth of 4 cm and the pots were irrigated to initiate the germination. Irrigation was carried out after sprouts emergence, 100 mL once every two days. Throughout the experiment, fertilization treatments and treatments against various pests were performed according to the growth protocol of the Israel Ministry of Agriculture Consultation Service (SAHAM). The experiments were performed in five repetitions for treatment (each repetition in a pot containing five sprouts). Emergence percentages were documented after eight days. After 42 days (five-leaves stage), the following indices were examined: general plant condition, the onset of symptoms; growth indices; and phenological stage. In addition, molecular quantitative real-time (qPCR) analysis was conducted to quantify the plant roots tissues' fungal DNA.

*2.6. Molecular Diagnosis*

2.6.1. Molecular Identification of the Endophytes

Mycelia from PDA-grown colonies were used for DNA extraction using the Master Pure Yeast DNA Purification Set Kit (Sigma, Rehovot, Israel). Molecular identification by PCR and sequencing was made by targeting the endophytes' small subunit ribosomal RNA gene, universal internal transcribed spacer (ITS and ITS4 primers, Table 2). PCR was done using the Rapidcycler (Idaho Technology, Salt Lake City, UT, USA) in a total volume of 20 μL per reaction: 1 μL of each primer (concentration of 20 μM), 10 μL of commercial reaction mixture RedTaq® ReadyMix (Sigma, Rehovot, Israel), 3 μL of template DNA and 5 μl autoclaved DDW. PCR conditions were 94 °C for 2 min, 30 rounds of 94 °C for 30 s, 55 °C for 30 s, 72 °C for one minute, and a finishing step of 72 °C for 5 min [7]. The PCR followed by cooling at 4 °C until recovery of the samples. The PCR products were sequenced by Hy Labs, Rehovot, Israel. Sequences were used to conduct a homology search against the GenBank using the BLASTN tool [42] (nucleotide blast on the NCBI website, https://blast.ncbi.nlm.nih.gov/Blast.cgi, accessed on 5 May 2021).

2.6.2. qPCR Diagnosis of *M. maydis* DNA in the Maize Plants

qPCR was performed on the plants' roots in the growth room experiment. The roots were washed with running tap water, then twice with sterile DDW, and cut into a section of about 2 cm. The weight of each repetition was adjusted to 0.7 g. DNA isolation and purification were done according to the protocol of Murray and Thompson (1980) [43], with slight modifications, as previously described [12].

The DNA samples were stored at −20 °C and used for the qPCR, as previously described [11]. This molecular method is based on a standard qPCR protocol used to detect mRNA (converted to cDNA) [44]. Instead, it was optimized to detect the DNA of the pathogen *M. maydis* using species-specific primers [45,46]. The A200a primers set was used for qPCR (sequences in Table 2). The housekeeping gene, COX, encoding the enzyme cytochrome c oxidase—the last enzyme in the cellular respiratory electron transport chain in the mitochondria, aimed at normalizing the *M. maydis* pathogen DNA [47]. This gene's amplification was done using the COX F/R primer set (Table 2). Calculation of the relative gene abundance was made according to the ΔCt model [48]. Similar efficacy was assumed for all samples. All amplifications were performed in triplicate.

**Table 2.** Primers used in this study.

| Pairs | Primer | Sequence | Uses | Amplification | References |
|---|---|---|---|---|---|
| Pair 1 | ITS1<br>ITS4 | 5′-TCCGTAGGTGAACCTGCGG-3′<br>5′-TCCTCCGCTTATTGATATGC-3′ | PCR target gene | Target region for the identification of fungi species | [49] |
| Pair 2 | A200a-for<br>A200a-rev | 5′-CCGACGCCTAAAATACAGGA-3′<br>5′-GGGCTTTTTAGGGCCTTTTT-3′ | qPCR target gene | 200 bp *M. maydis* species-specific fragment | [7] |
| Pair 3 | COX-F<br>COX-R | 5′-GTATGCCACGTCGCATTCCAGA-3′<br>5′-CAACTACGGATATATAAGRRCCRRAACTG-3′ | qPCR control | Cytochrome c oxidase (*Cox*) gene product | [47,50] |

The real-time PCR reactions were executed using the ABI PRISM 7900 HT Sequence Detection System (Applied Biosystems, CA, USA) and 384-well plates. The qPCR conditions were as follows: 5 μL total reaction volume was used per sample well—2 μL of DNA sample extract, 2.5 μL of iTaq™ Universal SYBR Green Supermix (Bio-Rad Laboratories Ltd., Hercules, CA, USA), and 0.25 μL of each of the forward and reverse primers (10 μM from each primer per well). The qPCR cycle program was as follows: Pre-cycle activation phase (1 min at 95 °C), denaturation (15 s at 95 °C) for 40 cycles, annealing and extension (30 s at 60 °C), and finalizing by melting curve analysis.

*2.7. Statistical Analysis*

A completely randomized statistical design was used to assess the endophytes treatment outcome for the symptoms in the growth room sprout infection. Data analysis followed by statistics was done using the JMP program, 15th edition, SAS Institute Inc., Cary, NC, USA. The one-way analysis of variance (ANOVA) was used with a significance threshold of $p < 0.05$. The ANOVA analysis followed by post hoc of the Student's *t*-test for each pair (without multiple comparisons correction).

## 3. Results

Endophytes are natural microflora that inhabit each maize seed and plant and may also be referred to as the plant's microbiome. These microorganisms' communities consist of several bacterial and fungal species, as can be seen in Figure 1. The current work focuses on isolating some of them and inspecting their bioprotective potential against the maize late wilt disease agent, *M. maydis*. In seeds, these endophytes may provide a protective shield against the pathogen's penetration and establishment at the beginning of plant development, before it can acquire new endophytes from the environment. Therefore, we isolated endophytes from maize grain (Figure 1), identified 11 of them (Table 3), and tested them in a direct-confront assay on media plates. The most successful antagonists to *M. maydis* were then used to enrich an LWD-susceptible cultivar's seeds and were challenged in a sprouts assay with highly infected soil.

In this study, 11 endophytes were isolated and identified (Table 3): 10 fungal species and one bacterial species, *B. subtilis*. The fungal species belonged to the genera *Trichoderma*, *Chaetomium*, *Penicillium*, *Fusarium*, *Alternaria*, and *Rhizopus*. Some are known as phytopathogens, so they possess complex rules in their host plant interactions (discussed in detail below). These endophytes were isolated from different maize cultivars (sweet and fodder) with various sensitivity to LWD. Therefore, these factors may be reflected in each of the maize genotypes' endophyte protective suit compositions.

For this reason, five species were selected for the follow-up sprout experiment based on their diverse source and success in the plates' confrontation assay. Although other endophyte species are also good candidates, they were not selected for several reasons. These include their potential to cause disease (*Fusarium proliferatum*, *Alternaria alternata*) because they possess human risk (*Rhizopus oryzae*) or because the same species was already selected (*Chaetomium cochliodes*).

The plate confrontation assay results (Figure 2) reveal that all identified endophytes (detailed in Table 3) can restrict *M. maydis* colony growth. Moreover, two species, *Trichoderma asperellum* (P1) and *Rhizopus oryzae* (J1), could grow above the *M. maydis* colony surface and cover its surface almost entirely in 6–7 days. Two of the selected isolates were

able to repress *M. maydis* through the secretion of antifungal metabolites into the growth medium: the bacteria *B. subtilis* (R2) and the fungus *Penicillium citrinum* (S7).

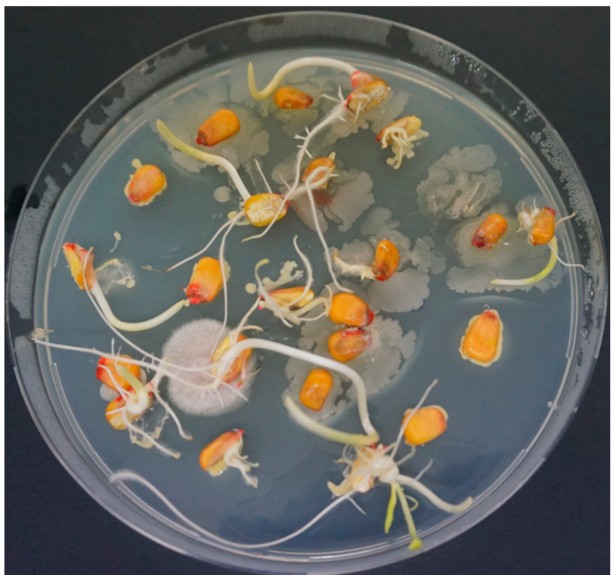

**Figure 1.** Isolation of endophytes from maize grains. Maize seeds were externally disinfected, cut lengthwise, and placed on a potato dextrose agar (PDA) substrate with the cutting surface turned downwards. Petri plates were incubated at 28 ± 1 °C in the dark for 2–3 days. At the end of this period, the fungal and bacterial endophytes that developed were isolated into new PDA plates and subjected to molecular identification.

**Table 3.** Endophytes identified in this study and plate confrontation assay results [1].

| Species | Designation | Systematic Classification | Maize Cultivar | NCBI Accession | NCBI Score | Confrontation Assay Winner | Tested in Sprouts |
|---|---|---|---|---|---|---|---|
| *Magnaporthiopsis maydis* | Hm-2 | Fungi | *Zea mays* | | | | |
| *Trichoderma asperellum* | P1 | Fungi | Prelude cv. | MF871569 | 99.66% | P1 | + |
| *Chaetomium subaffine* | M2 | Fungi | Megaton cv. | HM365247 | 100.00% | Antagonism | + |
| *Chaetomium cochliodes* | M4 | Fungi | Megaton cv. | MT520580 | 97.53% | Antagonism | + |
| *Penicillium citrinum* | S7 | Fungi | Simon cv. | MN046972 | 99.57% | Antagonism | + |
| *Bacillus subtilis* | R2 | Bacteria | Royalty cv. | MT415782 | 99.06% | Antagonism | + |
| *Chaetomium cochliodes* | M3 | Fungi | Megaton cv. | MN534819 | 99.80% | Antagonism | − |
| *Fusarium proliferatum* | S2 | Fungi | Simon cv. | MT563410 | 99.44% | Antagonism | − |
| *Fusarium proliferatum* | S9 | Fungi | Simon cv. | EU272509 | 99.25% | Antagonism | − |
| *Alternaria alternata* | R1 | Fungi | Royalty cv. | MK174979 | 99.46% | Antagonism | − |
| *Rhizopus oryzae* | J1 | Fungi | Jubilee cv. | MT603963 | 100.00% | J1 | − |
| *Fusarium proliferatum* | HT1 | Fungi | Hatai cv. | EU272509 | 98.89% | Antagonism | − |

[1] Confrontation assay results included the following possibilities: *M. maydis* or endophyte mycoparasitism (one of the microorganisms is growing above the colony surface of the other) and antagonism. In antagonism, none of them can extend above the other, and their growth was stopped at the meeting point with the other fungus, usually by producing a dark line. Antagonism may also result from the secretion of antipathogen compounds that blocks *M. maydis* colony growth.

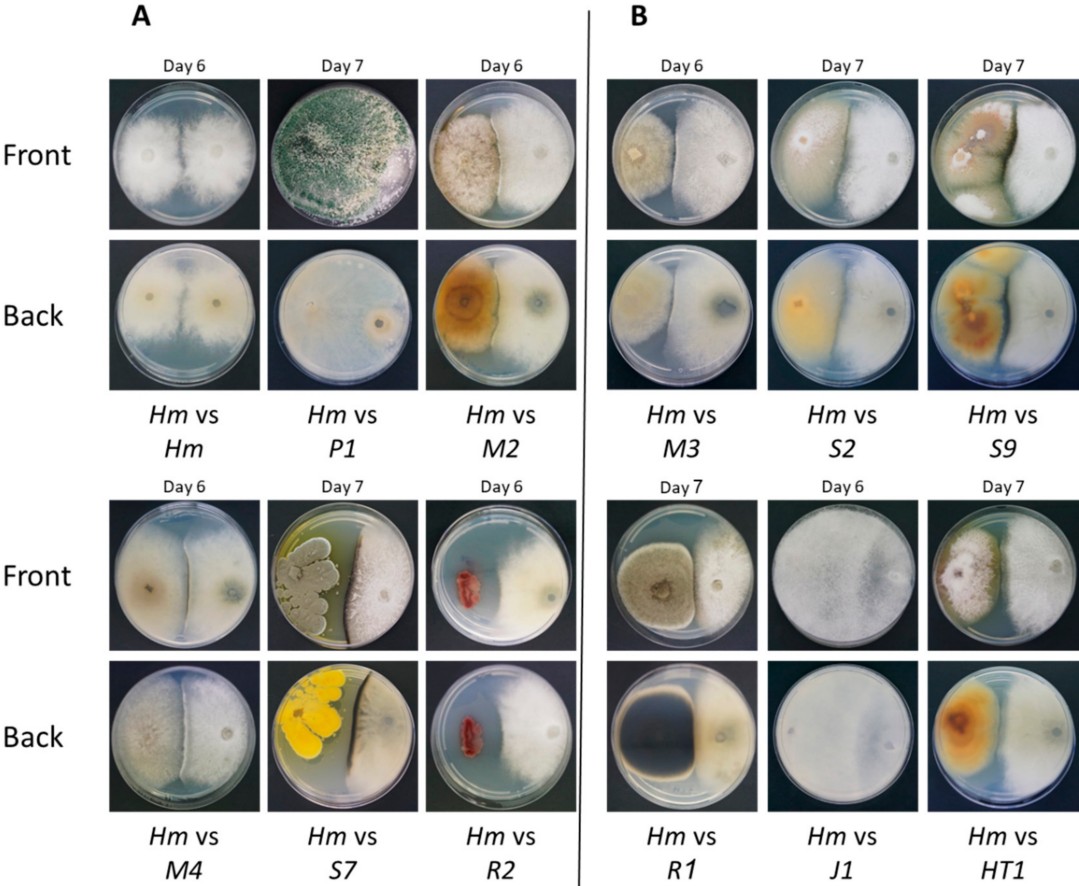

**Figure 2.** Plate confrontation assays. The PDA plate assay to identify interactions between *Magnaporthiopsis maydis* and different endophytes. The endophytes species tested are listed in Table 3. Upper left-most panel—*M. maydis* (*Mm*) was seeded on both sides of the petri dish. Other panels—the two fungi were planted opposite each other, endophytes species on the left, *M. maydis* on the right. (**A**). The endophytes selected for the subsequent sprout pathogenicity assay. (**B**). Endophytes species not selected for the sprouts assay. Each plate was photographed from both sides, front (upper panel) and rear (lower panel). Images were taken at 6–7 days of growth at 28 ± 1 °C in the dark as detailed above each figure.

The final step in evaluating the newly discovered endophytes for their biocontrol properties was to test them in *M. maydis* infected seedlings. The five selected species (see Table 3) were used to enrich maize seeds by inoculating the seeds with the endophytes for two days under optimal conditions. The treated seeds were then sown in heavily infected soil (with the LWD pathogen) and were grown for 42 days in a controlled environment. The experiment results reveal an interesting positive influence of two of the endophytes (Figure 3).

The emergence of the sprouts (evaluated 8 DAS) was similar, with no statistically significant differences, except for the *B. subtilis* (R2) treated plant that excels in this measure (reached 91% compared to 69% of the infected control, $p < 0.05$) (Figure 3A). On the other hand, the *T. asperellum* (P1) and *C. cochliodes* (M4) treatments had the lowest emergence rate (54%). Although this value was not statistically different from the healthy or infected control, this finding is interesting since P1 excelled (and M4 achieved good results) in improving all growth parameters 34 days later.

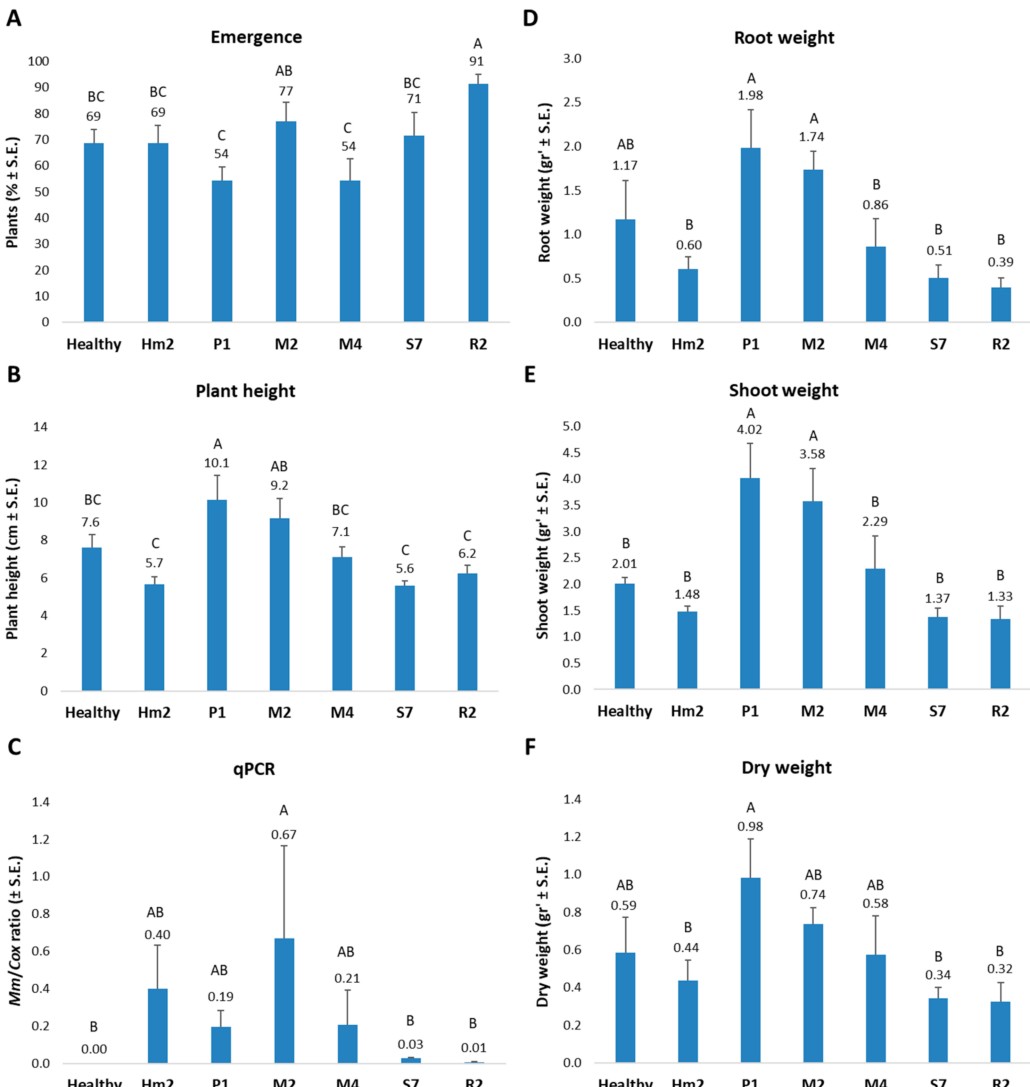

**Figure 3.** Seedlings pathogenicity assay. Symptoms evaluation was made for the maize late wilt-sensitive Prelude cv. after growing in soil infected with *M. maydis* in a growth chamber. The seeds were incubated with selected endophytes (Table 3) for two days under moderate humidity conditions prior to sowing. The control seeds underwent similar incubation without endophytes. The emergence percentage of the plants above the ground surface (**A**) was determined eight days after sowing. All other values (**B**–**F**) were documented after 42 days. The quantitative real-time PCR (qPCR) molecular method's values (**C**) are *M. maydis* relative DNA (*Mm*) abundance normalized to the cytochrome C oxidase (*Cox*) DNA. The experiment was performed in five repetitions. Error lines represent a standard error. Statistically significant (one-way ANOVA, $p < 0.05$) difference between the treatments is indicated by different letters (A–C).

At the experiment's end, 42 DAS, the plants in all the experiment groups reached the V5 phenological stage (fifth leaf appearance), except for the *M. maydis* infected, untreated control that achieved an average of 4.8 leaves per plant. In relation to the growth values measured at this age, the most significant improvement ($p < 0.05$) was of *T. asperellum* (P1) and *Chaetomium subaffine* (M2). These species' seed enrichment resulted in a significant elevation in plant height (1.8-fold for P1, Figure 3B), root and shoot fresh biomass (3.3- and 2.7-fold, respectively, for P1, Figure 3D,E), and plant dry weight (2.2-fold for P1, Figure 3F). The last was statistically significant only for P1. The *C. cochliodes* (M4) treatment resulted in improved values (but not statistically different) compared to the infected control. These results are intriguing since these three species (P1, M2, and M4) were isolated from the most LWD-susceptible maize cultivars (Prelude and Megaton, Table 1). In contrast, the

other two endophytes inspected in this assay, *Penicillium citrinum* (S7) and *B. subtilis* (R2) were isolated from resistant maize cultivars and had no beneficial impact on the growth parameters. Still, both S7 and R2 isolates had the most dramatic repression (13- and 40-fold, respectively) impact on *M. maydis* DNA levels inside the host plant's roots (Figure 3C). *T. asperellum* (P1) and *C. cochliodes* (M4) treatments resulted in only a ca. 2-fold decrease in *M. maydis* DNA compared to the infected control. At the same time, *C. subaffine* (M2) led to surprisingly high pathogen DNA levels.

## 4. Discussion

Late wilt of maize is a challenging disease that imposes a significant economic price in infected areas. The continued development of green solutions to control the fungal disease pathogen (*M. maydis*) efficiently for commercial maize production is an urgent need in Israel, Egypt, Spain, India, and other countries [27,51–53]. In recent years, devoted research efforts to identify and apply chemical pesticides against LWD in Israel produced encouraging results [11,19,20]. Commercial antifungal blends that contain Azoxystrobin can be used according to a timetable adjusted to *M. maydis* pathogenesis to shield susceptible maize hybrids, even in a highly infected area.

Still, intensive chemical treatment has several drawbacks in the short- and long run. In the short run, an intense chemical application may result in the appearance of fungicide resistance. Such cases have become gradually more common [54]. In the long run, phytoparasitic fungi's fungicide restraints may lead to human, animal, and environmental risks. Thus, restricting chemical fungicides use has become more and more essential and is currently a worldwide effort [21].

For these reasons, considerable research efforts in the past two decades were devoted to seeking alternative ways to control LWD. The mainstream of these efforts focuses on eco-friendly replacements of traditional chemical antifungal agents. These green methods included several research directions. First, a significant impact was achieved on the grain production and *M. maydis* presence in the field using the combined effect of minimum tillage and crop cover [55]. Second, a mixture of *Bacillus subtilis* and *Pseudomonas koreensis* resulted in siderophore production and antagonistic activity against *M. maydis*. Moreover, this combination prevented pre- and post-emergence damping-off and promoted the growth of greenhouse plants. The treatment also proved to be highly effective in reducing infection and increasing the yield [32]. Third, antagonistic phytopathogens, such as *Macrophomina phaseolina* noticeably improved the plants' health, while abolishing the severe late wilt symptoms in field conditions [18].

Agro-mechanical applications were also reported in varying degrees of maturity prior to field examination. Such methods demonstrated the beneficial impact of excessive irrigation (pot experiments conducted in an open-air enclosure) [52], applying plant extracts (*Lycium europaeum* under greenhouse conditions and at initial growth stage) [56], the use of plant growth hormones (in the lab, in vitro) [57], and inducing changes in root colonization by yeast (in vitro and under controlled greenhouse conditions), rhizobacteria and organic compounds (in the greenhouse and field) [31,58].

As in other fungal phytopathogens, *Trichoderma* species have received a central place in this global effort [28]. The current knowledge on LWD biological control using Trichoderma spp. was detailed in the introduction section. Whereas, some of these approaches were applied under controlled conditions, many were tested in field trials and produced encouraging results. Still, efforts by the scientific community to reveal the potential of such control strategies are continuing. Biological control has a significant advantage since the natural agent varied along with pathogen changes. A future integrative eco-friendly and chemical solution may combine the beneficial properties of both methods while drastically reducing the risks of chemical fungicides [21].

Seed treatments with bio-control formulations (*Bacillus subtilis*, *Bacillus pumilus*, *Pseudomonas fluorescens*, *Epicoccum nigrum*) were previously suggested for controlling maize LWD and tested in the field with encouraging results [59]. The seed treatments conducted

in two seasons, descendingly the *M. maydis* impact on pre-emergence damping-off, disease incidence, and crop yield. In another study, the influence of maize root colonization by microorganisms was also practical for this purpose [31]. For example, the rhizosphere actinomycetes *Streptomyces graminofaciens*, *S. rochei*, *S. annulatus*, *S. gibsonii*, *Candida glabrata*, *C. maltosa*, *C. slooffii*, and the fungi *Rhodotorula rubra* significantly reduced the growth of *M. maydis* in vitro and achieved positive results in seed dressing under controlled greenhouse conditions. Applying these species in the absence of the LWD pathogen significantly increased maize plant growth parameters [31].

The current study results contribute to this global effort and advance the use of beneficial maize endophytes as a bio-barrier and protective shield against the LWD fungus. Indeed, the identified partners in maize seeds natural microflora reported here can resist and even reduce the pathogen's development and spread inside the host roots. These antagonistic interactions improved the plants' immunity to the disease and strengthened them (increased their growth indices). In particular, *T. asperellum* (P1) and *Chaetomium subaffine* (M2) seeds enrichment was efficient and bioprotective at the sprouting phase. To the best of our knowledge, this is the first report on those two species as biocontrol agents of *M. maydis*. However, they were reported as bioprotective and growth prompters in other phytopathogens. In considering the integration of these species in new control applications, it appears that combining them may result in an improved outcome control method.

Whereas, *T. asperellum* (P1) and *C. subaffine* (M2) significantly improved the infected plants' growth indices, the pathogen DNA level detected from *C. subaffine* (M2) treatment is surprisingly high. Endophytes have more the one mode of action when confronting invasive pathogen [60]. Therefore, it is possible that *C. subaffine* is restricting the devastating pathogen activity but less influencing the pathogen spread. It should also be remembered that the relationship between symptom development and DNA quantity is not always in correlation [19]. This is especially true in resistant maize cultivars, at the early growth stage or when the disease is not severe. Hence, this intriguing question should be examined more deeply in future studies.

Interestingly, *P. citrinum* (S7) and *B. subtilis* (R2) repressed the pathogen in the plate assay through the secretion of antifungal metabolites in the plate confront assay. Indeed, both endophytes are well-studied and are known to produce a fungal inhibitory secretion and promote plant growth. The fungus *P. citrinum* has been reported as a common endophytic of cereal plants such as soybean and wheat. It produces mycotoxin citrinin and digesting enzymes (cellulase and endoglucanase, and xylulase), as well as plant growth hormones (summarized by [61]). *Bacillus* strains produce a great variety of antifungal compounds to suppress or kill fungal pathogens; thus, *Bacillus* species are often used to biocontrol plant diseases [62]. Among these compounds are non-ribosomal cyclic lipopeptides and volatile organic compounds (VOCs) with strong antifungal activities [63].

While *P. citrinum* (S7) and *B. subtilis* (R2) effectively limit *M. maydis* growth on plates, they were less effective in promoting the plants' growth under the *M. maydis* infection in the growth chamber. Still, they efficiently restrict the pathogen DNA in these seedlings. The classic triangle in phytopathology requires the interaction of a susceptible host, a virulent pathogen, and an environment favorable for disease development. In the plant confront assay, only two factors exist—the pathogen and the environment (abiotic—the growth medium and incubation conditions and biotic—the antagonists' endophytes). However, in the seedling test, the susceptible host is introduced, and this new factor can alter the results—the spread of the pathogen in the plant tissues and the resulting outbreak of the disease. This model is probably an oversimplification of the causes of the disease [64].

It should be remembered that many endophytes have more the one mode of action when confronting invasive pathogen. They can interact with it directly and inhibit its growth, but they can also induce the plant systemic defense response [60]. To support this, it is well-known that in vitro plate assays and even the seedlings pot assays having inconsistent ability to predict results in the field [11]. Still, these preliminary steps are necessary for ruling out ineffective pest-control treatments and choosing the ones with the

highest probability of success. Thus, the current study results encourage follow-up work that will test the most promising endophytes, in seed enrichment, under field conditions over a whole growth period.

A study of the non-pathogenic microbial community associated with maize could improve the management and yield production of maize crops under this threat. All maize plants in natural settings harbor seed-vectored endophytes that may influence the LWD resistance of maize cultivars. This study's results provide important support for endophytes' role as bioprotective microflora against *M. maydis*. It should be noted that resistance factors that differentiate susceptible LWD maize cultivars from resistance may result from differences in the endophytic communities harboring each cultivar.

One important endophytic bacterial species that was studied extensively is *B. subtilis* [41]. This bacteria genus is typical in seeds of different varieties of maize and may be transmitted vertically from one plant generation to the next as other endophytes [65]. This may imply the endophytes necessary for the survival of their host plant. These findings may support the idea that LWD-resistant maize genotypes acquired and inherited endophytes significantly impact their immunity to the *M. maydis* pathogen. Understanding the roles of how endophytic communities assemble and impact plant health could be an essential contribution to sustainable integrated agriculture.

## 5. Conclusions

The search for eco-friendly solutions to late wilt disease, besides the chemical options, is at the forefront of scientific efforts led by scientists from infected areas. The current study advances our understanding of the role of endophytes in acquired resistance to the disease. All 11 microorganism species isolated from maize grains and identified had an antagonistic effect against *M. maydis* in vitro (plate confrontation assay), while two excel in vivo (sprout assay). This first exploration of the endophytes' role regarding LWD suggests that the picture is much greater and more complex. For example, the maize cultivar resistance/susceptibility to the disease may be related to the endophytes colonize it. Indeed, the most successful species (*T. asperellum* (P1), *C. subaffine* (M2), and *C. cochliodes* (M4)), inspected in the pot assay here, were isolated from the most LWD-susceptible maize cultivars.

In contrast, the less beneficial endophytes, *P. citrinum* (S7) and *B. subtilis* (R2), were isolated from resistant maize cultivars. Still, these two less influencing isolates (regarding the plants' growth parameters) had the most drastic repression impact on *M. maydis* DNA levels inside the host plant's roots. Thus, these two species may have a significant influence in later growth stages. This aspect should be the focus of a follow-up study. The results encourage us to deepen and widen our understanding of this subject matter to reveal the true potential of the maize microbiome in the plant survival struggle against the pathogen. A better understanding of these interactions under natural conditions should help us understand (and influence) the long-term consequences of ruling out endophyte-based biocontrol.

**Author Contributions:** Conceptualization, O.D., D.R. and S.D.; methodology, O.D., D.R. and S.D.; Validation, O.D., D.R. and S.D.; formal analysis, O.D., D.R. and S.D.; investigation, O.D., D.R. and S.D.; resources, O.D.; Data curation, O.D., D.R. and S.D.; Writing—original D.R. preparation, O.D.; writing—review and editing, O.D. and S.D.; visualization, O.D.; supervision, O.D.; project administration, O.D.; funding acquisition, O.D. All authors have read and agreed to the published version of the manuscript.

**Funding:** This work was funded by a research grant from the Israel Falcha Workers Organization (2019). The APC was funded by Tel-Hai College, Upper Galilee, Tel-Hai, Israel.

**Institutional Review Board Statement:** Not applicable.

**Informed Consent Statement:** Not applicable.

**Data Availability Statement:** The datasets generated during and/or analyzed during the current study are available from the corresponding author on reasonable request.

**Acknowledgments:** We would like to thank Onn Rabinovitz (Israel Ministry of Agriculture) for his essential advice.

**Conflicts of Interest:** The authors declare no conflict of interest. The funders had no role in the design of the study; in the collection, analyses, or interpretation of data; in the writing of the manuscript, or in the decision to publish the results.

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
