# Peer review of "The Microflora of Maize Grains as a Biological Barrier against the Late Wilt Causal Agent, Magnaporthiopsis maydis"

_agronomy, doi:10.3390/agronomy11050965_

Round 1
Reviewer 1 Report
Very interesting work. Due to numerous restrictions on the use of chemical preparations in plant protection, introduced in many countries, it is necessary to strive to develop biological methods of plant protection. Therefore, the research is very necessary and up to date. The authors put a lot of work into the research carried out. All research is well planned. However, I have a lot of comments about the publication itself and how the results were presented.
line 50-51 the same sentence as in the abstract
In the introduction and later in the discussion, the works are cited, but there is no significant information that these works contain. For example, on lines 58-62 the authors report that research has already been carried out on biological methods to control M. maydis, but do not mention what species / endophytes were used for this. I think this is important information.
In the following parts of the manuscript you can be found this information but only in lines 74-83. Therefore, this paragraph should be placed immediately after line 62.
I also don't like the "most recently" used when quoting literature.
In general, biological methods are very poorly described, only two works are cited (22, 26).
line 169 - wheat seeds?
line 280 - it should be written what species have been selected.
S7 and R2 isolates were effective in limiting M. Maydis growth on plates. How the authors explain the lack of efficacy in seedling tests?
line 334-336 - I think this sentence is unnecessary
line 343-359 - In the discussion, the biological methods are described very generally. It is not possible to compare the results obtained by the authors with other similar studies. There is no information as to whether similar studies have been conducted on corn, with this pathogen and / or with these endophytes
line 360-362- why was it not stated what results were obtained ... there is nothing to discuss.
line 379-385- this is a conclusion rather than a discussion
line 395- 402 - these are not conclusions but an introduction
The weakest part of the manuscript is the discussion
Author Response
Responses to the reviewer 1 comments
We thank the reviewer for investing substantial efforts, which are undoubtedly contributing to this manuscript. The remarks and suggestions improved this paper’s scientific soundness and accurateness. Your contribution is greatly appreciated.
Line 50-51 the same sentence as in the abstract
The reviewer is correct. Thus, the sentence was rewritten and now sound: “As such, it may become pathogenic and cause disease in the right conditions.”
In the introduction and later in the Discussion, the works are cited, but there is no significant information that these works contain. For example, on lines 58-62, the authors report that research has already been carried out on biological methods to control M. maydis but do not mention what species/endophytes were used for this. I think this is important information.
We agree, and hence the following changes were incorporated in the text:
- The introduction and the discussion sections were rearranged, edited, and improved.
- The following paragraphs were rewritten, and additional information was added as requested by the reviewer (lines 69-100): “One of the research directions designed to address these challenges is biological control [28-32]. These methides include manipulating and strengthening beneficial microorganisms communities in the soil (for example, adding compost [29]) or direct applications of antagonistic bacteria and fungi or their secreted product. Such bacteria are Bacillus subtilis MF497446 and Pseudomonas koreensis (plant growth-promoting rhizobacteria [32]), and mixed strains of cyanobacteria known as Anabaena oryzae, Nostocmuscorum, and calcicolawere [29]. Instead, another approach was to use marine algae, and the cyanobacteria Anabaena oryzae extracts antifungal activates to targeting the LWD pathogen. These algae include Jania rubens, Corallina elongata, Laurencia obtusa, Gelidium crinale, Enteromorpha compressa, and Ulva fasciata.
LWD biological control using Trichoderma spp has also been demonstrated. Species in this genus can form mutualistic endophytic relationships with several plant species [33], while other species have been developed as biocontrol agents against fungal phytopathogens [34]. Previously, T. cutaneum reduced the incidence of M. maydis LWD of maize under greenhouse conditions by 89% compared to the control (from 94% to 11%) [31]. Similarly, T. harzianum applied in the field reduced the pre-emergence damping-off from 47% to 32% and increased the survival plants by 59% [35]. The use of T. viride alone, or even better with chitosan NPs, resulted in controlling late wilt in the greenhouse and field trials and improving the plants’ growth parameters [36]. Combined T. viride with the mycorrhizae led to a law effect on disease control.
It was also shown [28] that microalgae, Chlorella vulgaris extracts, with each of the Trichoderma species, T. virens, and T. koningii, were effective treatments against LWD under greenhouse and field conditions. The potential for applying Trichoderma-based methods against M. maydis in Israel has only lately been tested against the Israeli pathogen strains [37]. Examining eight marine and soil isolates of Trichoderma spp., known for high mycoparasitic potential, revealed that T. longibrachiatum isolates and of T. asperelloides has strong antagonistic activity against the Israeli pathogen isolate. These bioprotective agents were tested in a series of experiments in the laboratory and a growth room under controlled conditions until their final examination in pots under field conditions throughout a full growing season [37]. This green treatment has significantly improved the growth and yield indices to healthy plants’ levels and reduced pathogen DNA in the plants’ tissues by 98%.”
- The phrase “(Degani et al., unpublished data)” was replaced by a citation of a recently published work: Degani, O.; Dor, S. TrichodermaBiological Control to Protect Sensitive Maize Hybrids against Late Wilt Disease in the Field. Fungi 2021, 7, 315. https://doi.org/10.3390/jof7040315
- The following sentence and new reference were added (lines: 64-65): “Moreover, most resistant hybrids to LWD are low-yielding or have other undesirable agronomic characteristics [25].”
In the following parts of the manuscript, you can find this information but only in lines 74-83. Therefore, this paragraph should be placed immediately after line 62.
This is a correct remark. We reordered the paragraphs for more clarity, and they are now organized in the following logic:
- Lines 34-56: Late wilt disease (LWD) of maize.
- Lines 57-68: LWD control using the cultivation of disease-resistant corn genotypes.
- Lines 69-78: Utilizing beneficial microorganisms, direct applications, or secreted product.
- Lines 79-100: LWD biological control using Trichoderma
- Lines 101-113: The potential of maize endophytes as biocontrol agents.
- Line 114-121: Previous studies on fungal endophytes as a protective barrier against phytopathogens.
- Lines 122-132: Knowledge gaps and the current research goals.
I also don’t like the “most recently” used when quoting literature.
The word “recently” was removed from the text:
- Line 68: “… and was recently also reported in India [27].
- Line 89: “It was recently shown [28] that microalgae, Chlorella vulgaris extracts”
In general, biological methods are very poorly described; only two works are cited (22, 26).
As suggested by the reviewer, the entire paragraph regarding LWD biological control using Trichoderma spp. was rewritten, improved, and additional information was introduced. The section now summarizes 6 previous works (lines 79-100):
“LWD biological control using Trichoderma spp has also been demonstrated. Species in this genus can form mutualistic endophytic relationships with several plant species [33], while other species have been developed as biocontrol agents against fungal phytopathogens [34]. Previously, T. cutaneum reduced the incidence of M. maydis LWD of maize under greenhouse conditions by 89% compared to the control (from 94% to 11%) [31]. Similarly, T. harzianum applied in the field reduced the pre-emergence damping-off from 47% to 32% and increased the survival plants by 59% [35]. The use of T. viride alone, or even better with chitosan NPs, resulted in controlling late wilt in the greenhouse and field trials and improving the plants’ growth parameters [36]. Combined T. viride with the mycorrhizae led to a law effect on disease control.
It was also shown [28] that microalgae, Chlorella vulgaris extracts, with each of the Trichoderma species, T. virens, and T. koningii, were effective treatments against LWD under greenhouse and field conditions. The potential for applying Trichoderma-based methods against M. maydis in Israel has only lately been tested against the Israeli pathogen strains [37]. Examining eight marine and soil isolates of Trichoderma spp., known for high mycoparasitic potential, revealed that T. longibrachiatum isolates and of T. asperelloides has strong antagonistic activity against the Israeli pathogen isolate. These bioprotective agents were tested in a series of experiments in the laboratory and a growth room under controlled conditions until their final examination in pots under field conditions throughout a full growing season [37]. This green treatment has significantly improved the growth and yield indices to healthy plants’ levels and reduced pathogen DNA in the plants’ tissues by 98%.”
Line 169 - wheat seeds?
Indeed, these sterilized and M. maydis colonized wheat grains were used here to disperse the pathogen in the soil, as described previously (Zeller et al., Plant Dis. 2002, 86, 373-378). The same method was used by others (Ortiz-Bustos et al., Eur J Plant Pathol. 2015, 144 (2), 383-397), and by us (see for example Degani et al., Plant Dis. 2019. 103 (2), 238-248).
The sentence was rewritten to better explain this (lines 185-187): “The inoculation method also consists of sterilized, and M. maydis colonized wheat grains were used here to disperse the pathogen in the soil, as described previously [23].”
Line 280 - it should be written what species have been selected.
The endophytes species are detailed in Table 3. We add this explanation to the text:
- lines 292-293: “The plate confrontation assay results (Figure 2) reveal that all identified endophytes (detailed in Table 3) can restrict maydis colony growth. “
- Lines 299-301: “The five selected species (sees Table 3) were used to enrich maize seeds by inoculating the seeds with the endophytes for two days under optimal conditions.”
S7 and R2 isolates were effective in limiting M. maydis growth on plates. How do the authors explain the lack of efficacy in seedling tests?
This is true and should be better explained. The following paragraphs in the Discussion were edited and expanded (lines 421-449):
“Interestingly, P. citrinum (S7) and B. subtilis (R2) repressed the pathogen in the plate assay through the secretion of antifungal metabolites in the plate confront assay. Indeed, both endophytes are well studied and are known to produce a fungal inhibitory secretion and promote plant growth. The fungus P. citrinum has been reported as a common endophytic of cereal plants such as soybean and wheat. It produces mycotoxin citrinin and digesting enzymes (cellulase and endoglucanase, and xylulase), as well as plant growth hormones (summarized by [61]). Bacillus strains produce a great variety of antifungal compounds to suppress or kill fungal pathogens; thus, Bacillus species are often used to biocontrol plant diseases [62]. Among these compounds are non-ribosomal cyclic lipopeptides and volatile organic compounds (VOCs) with strong antifungal activities [63].
While P. citrinum (S7) and B. subtilis (R2) effectively limit M. maydis growth on plates, they were less effective in promoting the plants’ growth under the M. maydis infection. Still, they efficiently restrict the pathogen DNA in the host plants. The classic triangle in phytopathology requires the interaction of a susceptible host, a virulent pathogen, and an environment favorable for disease development. In the plat confront assay, only two factors exist – the pathogen and the environment (abiotic – the growth medium and incubation conditions and biotic – the antagonists’ endophytes). However, in the seedling test, the susceptible host is introduced, and this new factor can alter the result – the spread of the pathogen in the plant tissues and the resulting outbreak of the disease. This model is probably an oversimplification of the causes of the disease [64].
It should be remembered that many endophytes have more the one mode of action when confronting invasive pathogen. They can interact with it directly and inhibit its growth, but they can also induce the plant systemic defense response [60]. To support this, it is well known that in vitro plate assays and even the seedlings pot assays, despite having inconsistent ability to predict results in the field [11]. Still, these preliminary steps are necessary for ruling out ineffective pest-control treatments and choosing the ones with the highest probability of success. Thus, the current study results encourage a follow-up work that will test the most promising endophytes, in seed enrichment, under field conditions over a whole growth period.“
Line 334-336 - I think this sentence is unnecessary
The sentence was deleted as suggested by the reviewer.
Line 343-359 - In the Discussion, the biological methods are described very generally. It is not possible to compare the results obtained by the authors with other similar studies.
We agree so the information was elaborate and mow detailed in two paragraphs in the Discussion (lines 363-379):
“These green methods included several research directions. First, a significant effect on the grain production and M. maydis presence in the field was achieved using the combined effect of minimum tillage and crop cover [55]. Second, a mixture of Bacillus subtilis and Pseudomonas koreensis resulted in siderophore production and antagonistic activity against M. maydis. Moreover, this combination prevented pre and post-emergence damping-off and promoted the growth of greenhouse plants. The treatment also proved to be highly effective in the field in reducing infection and increasing the yield [32]. Third, antagonistic phytopathogens such as Macrophomina phaseolina noticeably improve the plants’ health while abolished the severe late wilt symptoms in field conditions [18].
Agro-mechanical applications were also reported in varying degrees of maturity prior to field examination. Such methods demonstrated the beneficial impact of excessive irrigation (pot experiments conducted in an open-air enclosure) [52], applying plant extracts (Lycium europaeum under greenhouse conditions and at initial growth stage) [56], the use of plant growth hormones (in the lab, in vitro) [57], and inducing changes in root colonization by yeast (in vitro and under controlled greenhouse conditions), rhizobacteria and organic compounds (in the greenhouse and field) [31,58]. “
There is no information as to whether similar studies have been conducted on corn, with this pathogen and/or with these endophytes
The above two new paragraphs cover the current knowledge and review similar studies. The subsequent paragraph summarizes the use of Trichoderma species towards this end. Trichoderma Based control against the LWD pathogen was detailed in the introduction section (we added a sentence that states this, lines 381-382). Seed treatments and maize root colonization by microorganisms are described in a new paragraph (lines 389-399). To the best of our knowledge, this is the first report on T. asperellum (P1) and Chaetomium subaffine (M2) as biocontrol agents of M. maydis. However, they were reported as bioprotective and growth prompters in other phytopathogens (we added a sentence stating this, lines 406-409).
line 360-362- why was it not stated what results were obtained ... there is nothing to discuss.
We agree with the reviewer and added the information, lines 389-393: “Seed treatments with bio-control formulations (Bacillus subtilis, Bacillus pumilus, Pseudomonas fluorescens, Epicoccum nigrum) were previously suggested for controlling maize LWD and tested in the field with encouraging results [59]. The seed treatments conducted in two seasons, descendingly the M. maydis impact on pre-emergence damping-off, disease incidence, and crop yield.”
line 379-385- this is a conclusion rather than a discussion
The paragraph was moved to the conclusion section, as advised.
line 395- 402 - these are not conclusions but an introduction
The information was deleted from the text, as advised, and the conclusion section was edited to maintain the logic of the data and the supported explanation.
The weakest part of the manuscript is the Discussion
We have edited and improved the Discussion, according to the reviewer suggestions and remarks, as follow:
- The following two paragraphs were rewritten and expend (lines 361-379): ”For these reasons, considerable research efforts in the past two decades were devoted to seeking alternative ways to control LWD. The mainstream of these efforts focuses on eco-friendly replacements of traditional chemical antifungal agents. These green methods included several research directions. First, a significant effect on the grain production and maydis presence in the field was achieved using the combined effect of minimum tillage and crop cover [55]. Second, a mixture of Bacillus subtilis and Pseudomonas koreensis resulted in siderophore production and antagonistic activity against M. maydis. Moreover, this combination prevented pre and post-emergence damping-off and promoted the growth of greenhouse plants. The treatment also proved to be highly effective in the field in reducing infection and increasing the yield [32]. Third, antagonistic phytopathogens such as Macrophomina phaseolina noticeably improve the plants’ health while abolished the severe late wilt symptoms in field conditions [18].
Agro-mechanical applications were also reported in varying degrees of maturity prior to field examination. Such methods demonstrated the beneficial impact of excessive irrigation (pot experiments conducted in an open-air enclosure) [52], applying plant extracts (Lycium europaeum under greenhouse conditions and at initial growth stage) [56], the use of plant growth hormones (in the lab, in vitro) [57], and inducing changes in root colonization by yeast (in vitro and under controlled greenhouse conditions), rhizobacteria and organic compounds (in the greenhouse and field) [31,58]. “
- The following paragraph was added (lines 389-399): “Seed treatments with bio-control formulations (Bacillus subtilis, Bacillus pumilus, Pseudomonas fluorescens, Epicoccum nigrum) were previously suggested for controlling maize LWD and tested in the field with encouraging results [59]. The seed treatments conducted in two seasons, descendingly the maydis impact on pre-emergence damping-off, disease incidence, and crop yield. In another study, the influence of maize root colonization by microorganisms was also practical for this purpose [31]. For example, the rhizosphere actinomycetes Streptomyces graminofaciens, S. rochei, S. annulatus, S. gibsonii, Candida glabrata, C. maltosa, C. slooffii, and the fungi Rhodotorula rubra significantly reduced the growth of M. maydis in vitro and in seed dressing under controlled greenhouse conditions. Applying these species in the absence of the LWD pathogen significantly increased maize plant growth parameters [31].”
- The following paragraph was added (lines 412-420): Whereas T. asperellum(P1) and C. subaffine (M2) significantly improved the infected plants’ growth indices, the pathogen DNA level detected from C. subaffine (M2) treatment is surprisingly high. Endophytes have more the one mode of action when confronting invasive pathogen [60]. Therefore, it is possible that C. subaffine is restricting the devastating pathogen activity but less influencing the pathogen spread. It should also be remembered that the relationship between symptom development and DNA quantity is not always in correlation [19]. This is especially true in resistant maize cultivars, at the early growth stage or when the disease is not severe. Hence, this intriguing question should be examined more deeply in future studies.”
- The following paragraph was added (lines 421-430): “Interestingly, citrinum (S7) and B. subtilis (R2) repressed the pathogenin the plate assay through the secretion of antifungal metabolites in the plate confront assay. Indeed, both endophytes are well studied and are known to produce a fungal inhibitory secretion and promote plant growth. The fungus P. citrinum has been reported as a common endophytic of cereal plants such as soybean and wheat. It produces mycotoxin citrinin and digesting enzymes (cellulase and endoglucanase, and xylulase), as well as plant growth hormones (summarized by [61]). Bacillus strains produce a great variety of antifungal compounds to suppress or kill fungal pathogens; thus, Bacillus species are often used to biocontrol plant diseases [62]. Among these compounds are non-ribosomal cyclic lipopeptides and volatile organic compounds (VOCs) with strong antifungal activities [63]. “
- The following two paragraphs were added (lines 431-449): “While citrinum (S7) and B. subtilis (R2) effectively limit M. maydis growth on plates, they were less effective in promoting the plants’ growth under the M. maydis infection. Still, they efficiently restrict the pathogen DNA in the host plants. The classic triangle in phytopathology requires the interaction of a susceptible host, a virulent pathogen, and an environment favorable for disease development. In the plat confront assay, only two factors exist – the pathogen and the environment (abiotic – the growth medium and incubation conditions and biotic – the antagonists’ endophytes). However, in the seedling test, the susceptible host is introduced, and this new factor can alter the result – the spread of the pathogen in the plant tissues and the resulting outbreak of the disease. This model is probably an oversimplification of the causes of the disease [64].
It should be remembered that many endophytes have more the one mode of action when confronting invasive pathogen. They can interact with it directly and inhibit its growth, but they can also induce the plant systemic defense response [60]. To support this, it is well known that in vitro plate assays and even the seedlings pot assays, despite having inconsistent ability to predict results in the field [11]. Still, these preliminary steps are necessary for ruling out ineffective pest-control treatments and choosing the ones with the highest probability of success. Thus, the current study results encourage a follow-up work that will test the most promising endophytes, in seed enrichment, under field conditions over a whole growth period. “

Reviewer 2 Report
Summary:
This paper gave some background information of the maize late wilt disease, introduced the pathogen, impact on maize production, and the management strategy. Utilization of disease resistant corn varieties is environmentally friendly and cost-effective method, however, it requires constant effort to identify new varieties. The authors proposed biological control using endophytic fungi and bacteria isolated from some varieties of maize plants and conducted experiments to study the interaction between these endophytic fungi/bacteria and the causal agent of the late wilt disease.
Broad comments:
- It was mentioned that R2 and S7 repress maydis through secretion of antifungal metabolites. Any further study on the metabolites?
- Figure 3 giving information on seedling pathogenicity assay. Is there any information about how many seeds got from the tested plants? Or probably study which species give good yield.
- Trichoderma asperellum (P1) and Chaetomium subaffine (M2) significantly improved the infected plants’ growth indices, however, the pathogen DNA level detected from M2 treatment is surprising high. Could you explain?
Specific comments:
- Make sure the designation of tested species is consistent in table 3 and figure 3. “P1” in table 3, while “1P” in figure 3.
- In Figure 2, some photo shows 6 days of growth, some shows growth on 7 days. Why not showing photos on the same day?
- In 2.3 Isolation and identification of endophytes from maize grains, isolated endophytes were subjected to molecular identification. Can you give more information on how to identify them?
Author Response
Responses to the reviewer 2 comments
We would like to express our sincere appreciation to the reviewer for essential and helpful advice. The time and effort invested are greatly appreciated and certainly contributed to the manuscript and improved it. Thank you.
Broad comments:
- It was mentioned that R2 and S7 repress maydisthrough secretion of antifungal metabolites. Any further study on the metabolites?
Yes, indeed (thank you for this question). The following paragraph was added to the Discussion to elaborate and answer this question (lines 421-430): “Interestingly, P. citrinum (S7) and B. subtilis (R2) repressed the pathogen in the plate assay through the secretion of antifungal metabolites in the plate confront assay. Indeed, both endophytes are well studied and are known to produce a fungal inhibitory secretion and promote plant growth. The fungus P. citrinum has been reported as a common endophytic of cereal plants such as soybean and wheat. It produces mycotoxin citrinin and digesting enzymes (cellulase and endoglucanase, and xylulase), as well as plant growth hormones (summarized by [61]). Bacillus strains produce a great variety of antifungal compounds to suppress or kill fungal pathogens; thus, Bacillus species are often used to biocontrol plant diseases [62]. Among these compounds are non-ribosomal cyclic lipopeptides and volatile organic compounds (VOCs) with strong antifungal activities [63]. “
- Figure 3 giving information on seedling pathogenicity assay. Is there any information about how many seeds got from the tested plants? Or probably study which species give a good yield.
This assay was conducted in the seedling grow stage (up to 43 days from sowing, five-leaves stage). Thus, there is no information about how many seeds got from the tested plants. The experiment was conducted with one maize genotype - Prelude cv. (sweet maize from SRS Snowy River Seeds, Australia, supplied by Green 2000 Ltd., Israel) susceptible to late wilt disease. This cultivar is commonly grown in commercial fields in Israel, and yield of over 20 tons per hectare is considered normal in healthy fields (see Degani et al., (2018), Effective chemical protection against the maize late wilt causal agent, Harpophora maydis, in the field. PLoS ONE 13(12): e0208353. https://doi.org/10.1371/journal.pone.0208353).
- Trichoderma asperellum(P1) and Chaetomium subaffine (M2) significantly improved the infected plants’ growth indices, however, the pathogen DNA level detected from M2 treatment is surprisingly high. Could you explain?
This is indeed a good question that should be better addressed. The following paragraph in the Discussion was added (lines 412-420):
“Whereas T. asperellum (P1) and C. subaffine (M2) significantly improved the infected plants’ growth indices, the pathogen DNA level detected from C. subaffine (M2) treatment is surprisingly high. Endophytes have more the one mode of action when confronting invasive pathogen [60]. Therefore, it is possible that C. subaffine is restricting the devastating pathogen activity but less influencing the pathogen spread. It should also be remembered that the relationship between symptom development and DNA quantity is not always in correlation [19]. This is especially true in resistant maize cultivars, at the early growth stage or when the disease is not severe. Hence, this intriguing question should be examined more deeply in future studies.”
Specific comments:
- Make sure the designation of tested species is consistent in table 3 and figure 3. “P1” in table 3, while “1P” in figure 3.
The labels in Figure 3 were corrected as advised and are now consistent with the labels in table 3.
- In Figure 2, some photo shows 6 days of growth, some shows growth on 7 days. Why not showing photos on the same day?
Since each endophyte has an individual growth rate, we added additional growth day to slow-growing species that will enable them to confront M. maydis.
- In 2.3 Isolation and identification of endophytes from maize grains, isolated endophytes were subjected to molecular identification. Can you give more information on how to identify them?
The molecular identification of the endophytes is explained in detail in section 2.6.1. Briefly, each endophyte was isolated to a clean colony. Hyphae from each colony were used for DNA extraction using DNA Purification Set Kit (according to the manufacture instructions). A PCR was conducted to targeting the endophytes’ small subunit ribosomal RNA gene, universal internal transcribed spacer (using the ITS and ITS4 primers). The PCR products were sequenced, and the sequences were used to conduct a homology search against the GenBank (using the BLASTN tool).
